# ERAP/HLA-C and KIR Genetic Profile in Couples with Recurrent Implantation Failure

**DOI:** 10.3390/ijms232012518

**Published:** 2022-10-19

**Authors:** Karolina Piekarska, Paweł Radwan, Agnieszka Tarnowska, Michał Radwan, Jacek R. Wilczyński, Andrzej Malinowski, Izabela Nowak

**Affiliations:** 1Laboratory of Immunogenetics and Tissue Immunology, Department of Clinical Immunology, Ludwik Hirszfeld Institute of Immunology and Experimental Therapy, Polish Academy of Sciences, 53-114 Wrocław, Poland; 2Department of Reproductive Medicine, Gameta Hospital, 95-030 Rzgów, Poland; 3Department of Surgical and Oncological Gynecology, Medical University of Łódź, 93-513 Łódź, Poland; 4Department of Surgical, Endoscopic and Oncologic Gynecology, Polish Mothers’ Memorial Hospital Research Institute, 93-338 Łódź, Poland; 5Medical Centre Gynemed, 93-346 Łódź, Poland

**Keywords:** ERAP, KIR, HLA-C, in vitro fertilization, RIF, polymorphism

## Abstract

Proper embryo implantation depends on the tolerance of the maternal immune system to the fetus and its foreign paternal antigens. During implantation and early pregnancy, the dominant leukocytes in the uterus are uterine NK cells, expressing killer immunoglobulin-like receptors (KIR). KIRs recognize human leukocyte antigens (HLA-C) on the human trophoblast inherited from the father and mother. The antigenic peptides presented by the HLA are formed via their cleavage by endoplasmic reticulum aminopeptidases ERAP1 and ERAP2. The aim of this study was to assess the association of combined *KIR* genes and their *HLA-C* ligands, as well as *ERAP1* and *ERAP2* polymorphisms with recurrent implantation failure after in vitro fertilization (RIF). We tested 491 couples who underwent in vitro fertilization (IVF) and 322 fertile couples. Genotype CC rs27044 *ERAP1* in female with a male’s *HLA-C1C1* or *HLA-C1C2* protected from RIF (p/p_corr_. = 0.005/0.044, OR = 0.343; p/p_corr_. = 0.003/0.027, OR = 0.442, respectively). Genotype TT rs30187 *ERAP1* in female with a male’s *HLA-C1C2* genotype increased the risk of RIF. Summarizing, in the combination of female *ERAP1* and an *HLA-C* partner, the rs30187 C>T and rs27044 C>G polymorphisms play an important role in implantation failure.

## 1. Introduction

Implantation is considered successful when an intrauterine gestational sac is recognized during an ultrasound scan. Implantation failure may occur in the early stages of migration or attachment of the embryo to the uterus. Recurrent implantation failure (RIF) is an absence of pregnancy after minimum three failed in vitro fertilization embryo transfers (IVF-ETs), with four good quality embryos transferred in a woman under the age of 40 years [1,2,3]. Proper implantation depends on the tolerance of the maternal immune system to the fetus expressing foreign paternal antigens without losing the mother’s ability to fight infection. The dominant population of leukocytes in the uterus during implantation and early pregnancy is the population of uterine NK cells (uNK) characterized by the CD56^bright^CD16- phenotype. In normal pregnancy, uNK cells lack cytolytic activity, but they release cytokines, chemokines and growth factors that regulate remodeling of the spiral arteries and influence trophoblast growth and differentiation [4,5,6,7,8]. Uterine NK cells express their own repertoire of activating and inhibitory receptors including killer immunoglobulin-like receptors (KIRs). The ultimate effect of uNK cells on pregnancy depends on the balance between activating and inhibitory receptors [5,8].

*KIR* genes are highly polymorphic. *KIRs* are divided into two haplotypes: A and B. *KIR3DL3*, *KIR3DP1*, *KIR2DL4* and *KIR3DL2* belong to framework genes and they are present in both haplotypes in almost all people [9,10]. Haplotype A consists of genes encoding only inhibitory receptors and only one gene corresponding to the activating receptor KIR2DS4. However, 80% of Europeans have a deletion of 22 bp at exon 5 in the *KIR2DS4* gene. Therefore, there is no expression of this receptor on the cell surface [11,12]. Haplotype B is much more diverse in terms of the total number of genes and the number of activating *KIR* genes than haplotype A. Haplotype B contains at least one of the following genes: *KIR2DS1*, *KIR2DS2*, *KIR2DL2*, *KIR2DL5A*, *KIR2DL5B*, *KIR2DS3*, *KIR2DS5*, *KIR3DS1* [13]. The AA genotype means an individual with two copies of haplotype A and possessing *KIR2DL1, KIR2DL3, KIR2DS4* and *KIR3DL1.* Individuals with the Bx genotype (AB + BB) carry genes for inhibitory receptors as well as genes for activating receptors in various combinations. *KIR* haplotypes have two regions: centromere (Cen) and telomere (Tel). This division is the result of exchange activity at the recombination point, located centrally within the *KIR* gene cluster [10,13].

KIR receptors recognize HLA class I ligands [8]. HLA-C allotypes are present on the trophoblast as the only polymorphic HLA class I molecules, as well as non-classical HLA-G and HLA-E molecules. HLA antigens on the trophoblast play an important role in regulating the immune response at the fetomaternal interface [14,15]. HLA-C alleles are divided into two groups: C1 and C2, depending on the presence of asparagine (HLA-C C1) or lysine (HLA-C C2) at position 80 in the HLA-C α-domain [16].

The antigenic peptides presented by HLA are formed via their cleavage to the appropriate length by endoplasmic reticulum aminopeptidases ERAP1 and ERAP2. They trim polypeptides from the N-terminal of the amino acid, thus creating antigens of optimal length to bind to HLA class I molecules. ERAP1 preferentially trims peptides with 9–16 amino acid residues and generates peptides with mainly 8–9 amino acids, while ERAP2 prefers to cleave shorter peptides (7–8 amino acids) [17,18].

*ERAP1* and *ERAP2* human genes are located on chromosome 5. *ERAPs* are polymorphic, and single nucleotide polymorphisms (SNPs) can result in expression of a product with different amino acid changes and this in turn can alter the enzymatic activity and substrate binding [19,20]. Moreover, the biological functions of some SNPs of *ERAP1* can be deduced from their location in the ERAP1 structure. The rs2287987 polymorphism (alteration of methionine to valine at position 349—Met349Val) is located in the catalytic site and could have an impact on interactions with the substrate. The rs27044 (Gln730Glu) on the inner surface of the C-terminal cavity could have influence on substrate sequence or length specificity. At domain junctions there are rs26653 (Pro127Arg) and rs30187 (Arg528Lys), which could induce the conformation change between open and closed and therefore impact on specificity and enzymatic activity [18,21,22]. The rs26618 polymorphism (Ile276Met) may influence the efficiency of generating an epitope related HLA-C*05 [23]. The rs6861666 A>G is in linkage disequilibrium (LD) with rs75862629, which is found in the promoter of the *ERAP2* gene. It has been shown that the presence of the G allele in the polymorphic site rs75862629 A>G *ERAP2* significantly influenced the expression of both ERAP1 and ERAP2 [24].

*ERAP2* shows limited polymorphism. The rs2549782 allele polymorphism causes an amino acid substitution of lysine (Lys392) to asparagine (Asn392) near the catalytic center of the enzyme. It affects the activity and specificity of ERAP2. The Asn392 is almost never expressed in humans, because of its strong linkage with SNP rs2248374 A>G. The rs2248374-G allele codes a gene isoform which is degraded due to the stop codon in the nonsense mediated decay process. The A allele of this polymorphism encodes a full-length transcript. Therefore, the rs2248374 polymorphism exerts an impact on the expression level of ERAP2 [19,20,21].

The polymorphism of the *ERAP1* and *ERAP2* genes may affect the correct presentation of antigens to the respective receptors. The polymorphism of *KIR* genes influences the activity status of NK cells and therefore may play a role in shaping interactions with HLA-C molecules expressed by the embryo.

In our previous work, we examined *ERAP* polymorphisms in the context of *KIR* and *HLA-C* genes only in women with RIF [25]. The aim of this study was to evaluate the association of combined *KIR*, *HLA-C*, *ERAP* gene polymorphisms in women and their partners with susceptibility to RIF.

## 2. Results

In this study we analyzed gene frequencies in couples who underwent in vitro fertilization and compared them to fertile couples. Table 1 shows summarized results, which are statistically significant after applying Bonferroni correction. A large number of comparisons are presented in detail in Appendix A.

### 2.1. Frequency of a Female KIR and Male HLA-C Genotype Combinations

Figures in next seven subsections were prepared to facilitate the understanding of the different combinations of female genes and their partner’s considered in the analysis.

Analysis of the frequency of the combination of the female KIR and the male HLA-C gene, showed only weak differences between IVF patients and fertile control, which lost significance after Bonferroni correction (Figure 1, Appendix A). The strongest difference was found for the combination of female KIR Tel BB with male HLA-C C1C2, which was more common in the fertile control than in the RIF group (^f^p/p_corr._ = 0.015/ns, OR = 0.149), and also more prevalent among SIVF couples compared to the RIF (^g^p/p_corr._ = 0.027/ns, OR = 0.160). In the most heterozygous genetic arrangement, when a woman had KIR genes located in the centromeric region AB and telomeric region AB and her partner had HLA-C1C2, couples were more widespread in the SIVF group than in the RIF group (^j^p/p_corr._ = 0.032/ns, OR = 0.380). Therefore, this combination seems to be more favorable in obtaining and maintaining a pregnancy (Appendix A).

### 2.2. Combinations of a Female ERAP and Male HLA-C Genotypes

In the next step, we estimated the frequencies of a female ERAP and male HLA-C genes combinations among IVF and fertile couples (Figure 2). For two SNPs of ERAP1 (rs30187 and rs27044) we found statistical differences. We observed that couples in which a man carried at least one C2 allele in the HLAC gene and his female partner had the TT rs30187 ERAP1 genotype were exposed to fertility problems (for HLA-C2 men: ^a^p/p_corr._ = 0.032/ns, OR = 1.962 in IVF vs. control; ^b^p/p_corr._ = 0.041/ns, OR = 2.062 in RIF vs. control; for men HLA-C1C2: ^g^p/p_corr._ = 0.006/0.050, OR = 2.843 in IVF vs. control; ^h^p/p_corr._ = 0.010/ns, OR = 2.907 in RIF vs. control; ^i^p/p_corr._ = 0.010/ns, OR = 3.290 in SIVF vs. control) (Table 1, Appendix A).

The combination of the CC ERAP1 rs27044 genotype in women with the HLA-C2 of their partners occurred more often in the fertile control group than in IVF, RIF, SIVF couples (^k^p/p_corr._ = 0.016/ns, OR = 0.647; ^l^p/p_corr._ = 0.031/ns, OR = 0.638; ^m^p/p_corr._ = 0.009/ns, OR = 0.534, respectively; Appendix A). Women positive for the rs27044 CC genotype and their partners with the HLA-C1C2 genotype, were also more often in the control group than in IVF, RIF and SIVF couples (^s^p/p_corr._ = 0.011/ns, OR = 0.591; ^t^p/p_corr._ = 0.034/ns, OR = 0.592; ^u^p/p_corr._ = 0.003/0.027, OR = 0.442, respectively; Table 1 and Appendix A). Additionally, the combination of female CC in rs27044 with male HLA-C1C1 protected from RIF, because the difference in comparison of RIF and SIVF couples was statistically significant even after correction (^p^p/p_corr._ = 0.005/0.044, OR = 0.343; Table 1 and Appendix A). On the other hand, women carrying the CG genotype in rs27044 along with HLA-C2 male partners, were overrepresented in IVF and SIVF group compared to the fertile control (^n^p/p_corr._ = 0.043/ns, OR = 1.472; ^o^p/p_corr._ = 0.015/ns, OR = 1.821, respectively; Appendix A).

Women with the CT genotype in rs2287987 together with their partners carrying at least one HLA-C2 allele, had a lower risk of infertility than couples without this genotype configuration—this combination was more prevalent in the fertile group in comparison to the IVF group (^x^p/p_corr._ = 0.049/ns, OR = 0.690, Appendix A).

### 2.3. Combinations of Female ERAP/KIR with Male HLA-C Genotypes

We analyzed the frequencies of female ERAP/KIR genotypes with male HLA-C combinations (Figure 3). We found that in women whose partners were heterozygous HLA-C1C2, female CC genotype in rs27044 ERAP1 with the combination of female KIR Bx genotype had a protective effect from RIF—the percentage of couples carrying this combination was more frequent in the fertile control than RIF (^k^p/p_corr._ = 0.004/0.037, OR = 0.428; Table 1) and IVF group (^j^p/p_corr._ = 0.034/ns, OR = 0.570) (Appendix A). The female CG rs27044 ERAP1 genotype had the opposite effect—this combination with the female KIR Bx genotype and male HLA-C1C2 was more prevalent in RIF (^m^p/p_corr._ = 0.006/0.050, OR = 2.313; Table 1 and Appendix A) and IVF couples (^l^p/p_corr._ = 0.043/ns, OR = 1.701) than in controls. Other results were weakly statistically significant (Appendix A).

### 2.4. Combinations of Female KIR/HLA-C with Male ERAP Genotypes

The analysis of the frequency of the female KIR/HLA-C and the male ERAP showed no strong differences between the analyzed groups—almost all the results lost significance after the Bonferroni correction (Figure 4). Only in the case where the male CT rs30187 ERAP1 genotype was in combination with female HLA-C2C2 and KIR AA, significantly more couples were in the RIF group than in the SIVF group (^c^p/p_corr._ = 0.005/0.049, OR = 24.000; Table 1 and Appendix A). Men positive for rs30187 TT along with female HLA-C1 and KIR AA genotypes were protected from RIF (^a^p/p_corr._ = 0.043/ns, OR = 0.210; ^b^p/p_corr._ = 0.025/ns, OR = 0.154, respectively; Appendix A). A similar effect was shown, when the genotype GG rs27044 ERAP1 in men was in combination with HLA-C1/KIR AA in women (RIF vs. SIVF: ^d^p/p_corr._ = 0.011/ns, OR = 0.087) and male genotype CC rs26653 ERAP1 was in combination with female HLA-C1/KIR AA (RIF vs. fertile: ^g^p/p_corr._ = 0.015/ns, OR = 0.114; RIF vs. SIVF: ^h^p/p_corr._ = 0.020/ns, OR = 0.098).

It was also noted that men with the genotype AG in rs2248374, whose female partners had the HLA-C1C1/KIR Bx combination, were more often in the IVF group than in the control group (^t^p/p_corr._ = 0.035/ns, OR = 1.964). The opposite effect was observed in the GG rs2248374 genotype in men together with their female partners carrying the HLA-C1C1/KIR Bx combination. Those couples were more prevalent in the fertile group than in the IVF group (^u^p/p_corr._ = 0.020/ns, OR = 0.426) and RIF (^w^p/p_corr._ = 0.030/ns, OR = 0.397; Appendix A).

### 2.5. Combinations of Female KIR with Male HLA-C and ERAP Genotypes

The frequency of the combination of the female KIR genotype with the male HLA-C and ERAP was similar in the studied groups, hence the obtained differences mostly lost their significance after Bonferroni correction (Figure 5, Appendix A). However, we found that the male genotype GG in rs27044 ERAP1 in combination with HLA-C1 showed a protective effect when female partners carried KIR AA (RIF vs. SIVF: ^c^p/p_corr._ = 0.005/0.031, OR = 0.114, Table 1 and Appendix A). This effect was maintained, when men were HLA-C1C1 homozygous, but the significance was lost after Bonferroni correction (^d^p/p_corr._ = 0.046/ns, OR = 0.000). Analysis of the comparison of RIF and SIVF couples revealed that more couples in RIF than in SIVF had the male CC ERAP1 rs27044/HLA-C2C2 and female KIR Bx combination (^e^p/p_corr._ = 0.014/ns, OR = 7.501). Conversely, the CG genotype in rs27044 and HLA-C2C2 in men with the female KIR Bx combination was protective against RIF (RIF vs. SIVF: ^g^p/p_corr._ = 0.004/0.033, OR = 0.096; Table 1 and Appendix A).

### 2.6. Combinations of Female ERAP/HLA-C with Male HLA-C Genotypes

We also analyzed triple combinations of female ERAP and HLA-C with male HLA-C genotypes (Figure 6). Due to the low significance of results, they were included in the Appendix A. The exception was the case where the woman was TT homozygous for rs30187 ERAP1 and simultaneously a carrier of HLA-C2, while her partner had HLA-C1 allele. Such couples were more frequent in the RIF group than in the fertile control group (^a^p/p_corr._ = 0.008/ns, OR = 2.906; Appendix A). When the partner had HLA-C2 in this combination, we also observed a predisposing effect to RIF, but it was weaker (^b^p/p_corr._ = 0.028/ns, OR = 2.537; Appendix A).

### 2.7. Combinations of Male ERAP/HLA-C with Female Partner’s HLA-C Genotypes

As in the previous section, triple combinations were tested by considering the HLA-C genes of women and partners together, but the ERAP genes were of paternal origin (Figure 7). Results are included in Appendix A. The CC genotype in rs30187 ERAP1 together with the HLA-C2 in males in combination with female HLA-C1, showed a weak predisposing effect to RIF (^b^p/p_corr._ = 0.042/ns, OR = 1.608). The same configuration of male and female HLA-C genotypes in combination with the male CC rs27044 ERAP1 genotype was also exposed couples to the risk of implantation failure (RIF vs. fertile: ^d^p/p_corr._ = 0.030/ns, OR = 1.655; RIF vs. SIVF: ^e^p/p_corr._ = 0.020/ns, OR = 1.914; Appendix A).

## 3. Discussion

This study shows for the first time that some genetic combinations of *ERAP*, *KIR*, *HLA-C* in women and their partners may predispose to RIF and infertility. The *ERAP1* and *ERAP2* genes encode endoplasmic reticulum aminopeptidases, which play an important role in the proper cleavage of peptides that generate structurally stable antigens binding to HLA class I. The polymorphisms of these genes impact the various activities of aminopeptidases, therefore, they influence the formation of HLA-antigen complexes. The inability to form the correct complexes can result in a lack of an immune response by NK cells or CD8+ lymphocytes. Thus, *ERAP1* and *ERAP2* polymorphisms may affect the KIR-HLA-C interaction at the maternal-fetal interface.

We found that *KIR* genes in the B telomeric region in women in combination with her partner’s *HLA-C1C2* were more frequent in the fertile control and SIVF groups than in the RIF group. Moreover, the combination in which the woman had *KIR* Cen BB/Tel BB and her partner *HLA-C1C2* was not present in the RIF group, while about 71% of fertile couples possessed it. In addition, having a heterozygous *KIR* Cen AB/Tel AB by a woman, when her partner carrying heterozygous *HLA-C1C2* act conductively to achieving pregnancy after IVF (Appendix A).

The maternal *KIR* AA genotype is considered as a risk factor, when the fetus carries the *HLA-C2* allele as it can severely inhibit the secretion of cytokines and growth factors from NK cells that are necessary for the successful development of an embryo. The KIR2DL1 receptor, the gene of which belongs to *KIR* AA, binds to the HLA-C2 epitope and their interaction is stronger than KIR2DL2 or KIR2DL3 with HLA-C1 [26,27]. *KIR* A haplotype means there are no genes coding activating receptors including *KIR2DS1*, which is present in the B telomeric region. Haplotype B is characterized mostly by the presence of genes for activating *KIR*. Therefore, we observe the opposite situation in women who possess the *KIR* Bx genotype (*KIR* AB or BB). Even if the embryo inherits *HLA-C2* from the father, the strong inhibition that follows from the interaction of KIR2DL1 and HLA-C2 will be counterbalanced by activating signals from the activating KIR receptors, resulting in better trophoblast invasion. Instead of KIR2DL1, KIR2DS1 (which is quite common in the Polish population—about 50%) can bind to the HLA-C2 molecule [28]. However, the inhibitory receptor has a greater affinity for the ligand than KIR2DS1 depending on which receptor is expressed on the cell surface [29]. In addition to KIR receptors, there are also other receptors on the surface of NK cells with an activating and inhibiting function, such as: CD94/NKG2, NKG2D and natural cytotoxicity receptors—NCRs (NKp30, NKp46, NKp44) [30].

Xiong et al. (2013) showed that GM-CSF (granulocyte-macrophage colony-stimulating factor) alongside IFN-γ and the chemokines XCL2 and CCL3L3, which promote trophoblast invasion, were secreted by KIR2DS1-positive uterine NK cells and its interaction on target cells with the HLA-C2 molecule [31]. This interaction enhanced the migration of primary trophoblast cells, therefore these studies emphasize how beneficial the KIR2DS1-HLA-C2 interaction is for placenta formation. Hiby et al. (2010) reported that the presence of the *KIR2DS1* gene in haplotype B provides protection against pregnancy disorders, such as recurrent miscarriages, preeclampsia or intrauterine fetal growth restriction [32]. Additionally, they confirmed that a higher frequency of *KIR* AA genotype in women, in comparison to the control group, was associated with pregnancy disorders. Moreover, in all study groups, both mother and fetus had a higher frequency of *HLA-C2* compared to the control group [32]. In our previous study, we showed that the combination of *KIR* genes from the telomeric region A and *HLA-C2C2* in women was associated with infertility and RIF, which confirms our considerations [25]. Additionally, this work demonstrated the role of the *KIR*/*HLA-C* gene system with the *ERAP* genes in implantation failure. In that previous analysis, among all tested *ERAP* polymorphisms, rs26653 and rs26618 had the greatest influence on infertility and RIF in women carrying *HLA-C2* [25].

In this research we showed for the first time the possible interactions between woman and her partner within the combinations of genes *ERAP*/*HLA-C*/*KIR*. Figure 8 shows significant genetic interactions between the woman and her partner that could potentially lead to implantation failure or be protective and conducive to pregnancy. We found that the TT rs30187 *ERAP1* genotype in women predisposed to infertility, when the partner had *HLA-C1C2* (Table 1, Appendix A and Figure 8A). The polymorphism rs30187 C>T causes the replacement of arginine with lysine at position 528 (Arg528Lys), which results in a change between the closed and open conformation of the ERAP1 and thus may affect its specificity. The TT genotype of rs30187 *ERAP1* (variant coding lysine) encodes an enzyme which has higher activity and expression [33]. In turn, higher enzyme activity, expressed by excessive trimming, may result in the destruction of the peptides presented by HLA-C. The TT genotype was also associated with recurrent miscarriages after natural fertilization, as it was shown in our previous published study [34]. Predisposition to miscarriage was noted when a woman had TT rs30187 *ERAP1* genotype and the *HLA-C2* allele or *HLA-C1C2* genotype [34]. In this study the predisposition to infertility was present in the configuration of a woman’s TT rs30187 *ERAP1* and *HLA-C1C2* of her partner. A similar effect was observable in the case when a male had the CT rs30187 *ERAP1* genotype and his female partner had the *KIR* AA/*HLA-C2C2* combination (Table 1, Appendix A and Figure 8D). From these results, it can be concluded, that the T allele in rs30187 of *ERAP1* polymorphism, whether maternal or paternal, is unfavorable to achieve and maintain pregnancy.

ERAPs also play a key role in blood pressure regulation, due to their involvement in the local renin-angiotensin pathway. ERAP1 cleaves bioactive hormone angiotensin II to angiotensin III, in turn ERAP2 cleaves angiotensin III to angiotensin IV and kallidin to bradykinin [35]. Angiotensin III and kallidin are involved in regulating the extension and constriction of blood vessels. Abnormalities in the processing of angiotensin and kallidin can contribute to high blood pressure. Yamamoto et al. (2002) found that the variant Arg528 rs30187 (C allele) of *ERAP1* had less activity than the Lys528 (T allele) form [36]. The C allele causes that the enzyme cleaves angiotensin II with 60% reduced efficiency [37]. This in turn can lead to high blood pressure and hypertension, because of insufficient inactivation of angiotensin II and/or bradykinin formation [36]. Ferreira et al. (2021) showed, that angiotensin II level in plasma of pregnant women did not differ among rs30187 genotypes groups, which also disproves the hypothesis that women carrying the C allele have increased levels of angiotensin II in plasma which results in hypertension [38]. The authors explain this with the fact that both variants Arg528 and Lys528 are associated with bad outcomes. Lys528 was strongly associated with susceptibility to ankylosing spondylitis and other autoimmune diseases [22,39]. It is very possible that *ERAP* genes are subject to balancing selection, which allows the favoritism and better adaptive qualities of heterozygous than homozygous variants [38]. However, it does not work for the men tested in this study. Heterozygous men were prone to infertility with their female partners. Perhaps the C allele causing hypertension could be also associated with an increase in reactive oxygen species. This can lead to nuclear and mitochondrial DNA damage in sperm, telomere shortening, epigenetic changes and Y chromosome microdeletions [40,41].

The rs27044 C>G *ERAP1* polymorphism leads to the change of the amino acid—glutamine with glutamic acid at position 730 (Gln730Glu)—in the catalytic domain IV of ERAP1 and may influence the substrate length preference [22]. Our results indicate that allele C in rs27044 can reduce risk of infertility in women, if an embryo inherits alleles *HLA-C1* or *HLA-C2* from the father (Table 1, Appendix A and Figure 8B). Additionally, we observed the same effect, when a woman had the *KIR* Bx genotype (Table 1, Appendix A and Figure 8C). On the other hand, predisposition to RIF was observed in women carrying the CG rs27044 *ERAP1* and *KIR* Bx genotypes together with the male *HLA-C1C2* (Table 1, Appendix A).

Women, regardless of their *KIR* genotype (*KIR* Bx or *KIR* AA), had a greater chance of successful embryo transfer after in vitro fertilization, resulting in pregnancy and childbirth, when their partners possessed at least one G allele in the rs27044 C>G *ERAP1* (GG or CG genotype) and *HLA-C1* or *HLA-C2C2*, (Table 1, Appendix A and Figure 8E,F). Stamogiannos et al. (2015) found that the amino acid changes caused by polymorphism in the rs27044 of *ERAP1*, due to its localization, lead to increased enzyme preference for smaller peptides [33]. Position 730 of ERAP1 is located in the inner cavity of the ERAP1 enzyme that binds peptide substrates. This may directly interfere with the ability to bind longer peptides and activate their trimming. Binding to longer peptides is regarded to be necessary to induce a conformational change in which the catalytic moiety is moved towards the active site of the enzyme. Then, the longer peptides are effectively trimmed and the shorter peptides (8–9 amino acid residues) are trimmed more slowly. In particular, the rs27044G variant (amino acid at position 730 with glutamic acid—730Glu) is of importance. This amino acid carries a negative charge, that can affect the local electrostatic potential, making it difficult to accommodate the negatively charged C-terminus of long peptides. In turn, over-trimming of shorter peptides can lead to their destruction [33,42]. This could be consistent with our results obtained in the context of *ERAP1* rs27044 in women (Table 1).

Additional effects were noticed when an embryo inherits the G allele from its father. In a situation when the trophoblast expresses HLA-C1 molecules inherited from the father and the mother’s NK cells express KIR receptors encoded within *KIR* A haplotype, the interaction can only occur through KIR2DL3 (Table 1, Appendix A and Figure 8E). Activity changes of the enzyme ERAP1 could result in the production of peptides that may be incorrectly presented by HLA-C1—ligand to the KIR2DL3. Consequently, uterine NK cells will not be inhibited, which is advantageous in terms of pregnancy outcome. When the trophoblast expresses HLA-C2 inherited from the father and the mother’s NK cells express KIR receptors encoded within haplotype B and A (Bx genotype), the interaction can be due to KIR2DL1 or KIR2DS1, depending on whether the expression of inhibitory or activating KIR receptors prevails. Furthermore, the presence of paternal origin CG genotype rs27044 ensures partially proper enzyme function (Figure 8F).

The limitation of our studies is that no quadruple combinations were analyzed, where female *ERAP/HLA-C* with male *ERAP/HLA-C* comparisons can be undertaken. Furthermore, the numbers in study groups should be increased so that the analysis gives meaningful results. There were also no significant results related with the *ERAP2* genotypes. Moreover, we did not measure the level of ERAPs in the plasma of men. As we showed in our previous research, the concentration of ERAP2 in women’s plasma was associated with the pregnancy, and high levels of ERAP2 could be related with the miscarriage [25]. Unlike *ERAP1*, the *ERAP2* gene shows limited polymorphism. There are two main allotypes due to the rs2248374 A>G polymorphism. The A allele encodes a full-length and full-functionality transcript (ERAP2-A isoform). In contrast, the G allele encodes the ERAP2-B isoform, which is degraded due to premature stop codon in nonsense-mediated decay. Therefore, individuals with genotype AA or AG expression full length *ERAP2*, unlike individuals with GG genotype (25% of the population) [20,21,43]. However, it was observed, that despite the nonsense-mediated decay process, the G allele can lead to the formation of short isoforms Iso3 and Iso4 which have been expressed after viral infection [21,43,44]. Since ERAP2 also regulates blood pressure and renin-angiotensin system, it would be worth investigating the levels of ERAP2 in men’s plasma and correlate it with genotypes of *ERAP2*.

In conclusion, individual genetic variants of *ERAP* in combination with *HLA-C* or *KIR* may impact on the activity of ERAP enzymes and incorrect trimming of antigenic peptides, which prevents the proper recognition of the HLA on the embryo by the mother’s KIR receptors. *ERAP1* rs30187 and rs27044 polymorphisms in combination with *HLA-C* and *KIR* genes seems to play the greatest role in susceptibility or protection from RIF in couples undergoing in vitro fertilization. As we showed, some combinations proved to be important when we considered male *ERAP*. Therefore, the immunogenetic role of the father should not be overlooked in implantation failure.

## 4. Materials and Methods

### 4.1. Study Design

We studied four hundred ninety-one women and their partners, who underwent IVF-ETs with a total number of transferred embryos amounting 1834. Patients were qualified at the Gameta Assisted Reproduction Clinic in Rzgów, Centre certified by the European Society for Human Reproduction and Embryology (ESHRE ART Centre Certification for good clinical practice) and at the Department of Surgical, Endoscopic and Oncologic Gynecology, Department of Gynecology and Gynecologic Oncology, Polish Mothers’ Memorial Hospital Research Institute. Among patients, we distinguished three subgroups. Two hundred seventy-eight were RIF (mean of five unsuccessful transfers). A group of 161 couples constituted the SIVF patients (successful pregnancy after IVF-ET). Most women in the SIVF group became pregnant after the second embryo transfer and they had a clinically advanced pregnancy or gave birth after IVF-ET. Fifty-two couples could not be qualified for RIF or SIVF because they had 1–3 embryo transfers and most of them experienced a miscarriage. The control fertile group consisted of 322 fertile couples. Women in this group become pregnant after natural conception and they had at least one healthy-born child. They did not have any miscarriage or immunological and endocrinological diseases. The control group was qualified from the 1st Department of Obstetrics and Gynecology, Medical University of Warsaw, in the years 2006–2014 and from the Institute of Immunology and Experimental Therapy of the Polish Academy of Sciences in 2018–2020. All tested couples were of Polish origin. The mean age of IVF patients and their partners was significantly different from the mean age of the fertile control (*p* < 0.0001, *p* < 0.0001, respectively). The clinical characteristic of patients and fertile couples is presented in Table 2. All studies were carried out after obtaining the approval of the Ethics Committee of the Polish Mothers’ Memorial Hospital–Research Institute in Łódź (Poland). Necessary consent was obtained from all individual participants.

### 4.2. DNA Preparation and Genotyping

Genomic DNA isolation from venous blood was performed according to manufacturer’s protocol, using the Invisorb Spin Blood Midi Kit (Invitek, Berlin, Germany) or QIAamp DNA Mini Kit (Qiagen, Hilden, Germany). We used KIR Ready Gene kits (Inno-train Diagnostics, Kronberg, Germany) according to the manufacturer’s instruction or multiplex PCR described elsewhere [45,46] to determine the genotypes of *KIR*. *KIR* AA genotype means the presence of *KIR2DL1*, *KIR2DL3*, *KIR2DS4*, *KIR3DL1* and the absence of *KIR2DL5*, *KIR2DS1*, *KIR2DS2*, *KIR2DS3*, *KIR2DS5* and *KIR3DS1* which are characteristic for *KIR* Bx genotype. *KIR* genes are also divided according to the presence in the centromeric or telomeric part of the *KIR* gene cluster: CenA-*KIR2DL3*, CenB-*KIR2DL2* and *KIR2DS2*, TelA-*KIR3DL1* and *KIR2DS4*, TelB-*KIR2DS1* and *KIR3DS1*. To determine the *HLA-C* C1 and C2 allotypes, we used the PCR-SSP method described elsewhere [47]. *ERAP1* and *ERAP2* polymorphisms were examined by the TaqMan SNP Genotyping Assay (Applied Biosystems, Waltham, USA) as described previously [34]. Table 3 presents the characteristics of the studied *ERAP* polymorphisms.

### 4.3. Statistical Analysis

Frequencies of *KIR, HLA, ERAP* genotypes and their combinations were compared using the two-tailed Fisher’s exact test in the R package version 3.4.3. *p* values < 0.05 were considered statistically significant. The odds ratio (OR) and 95% confidence interval (95% CI) were estimated as the measure of effect size. We checked whether the frequencies of genotypes, both in patients and in the control group, were consistent with the Hardy-Weinberg equilibrium using the chi-square test with 1-degree-of-freedom (df). The Bonferroni correction was also used for multiple comparisons.

## 5. Conclusions

Individual genetic variants of *ERAP* in combination with *HLA-C* or *KIR* may impact on the activity of ERAP enzymes and incorrect trimming of antigenic peptides, which prevents the proper recognition of the HLA on the embryo by the mother’s KIR receptors. *ERAP1* rs30187 and rs27044 polymorphisms, in combination with *HLA-C* and *KIR* genes, seem to play the greatest role in susceptibility or protection from RIF in couples undergoing in vitro fertilization. As we showed, some combinations proved to be important when we considered male *ERAP*. Therefore, the immunogenetic role of the father should not be overlooked in implantation failure.

## Figures and Tables

**Figure 1 ijms-23-12518-f001:**
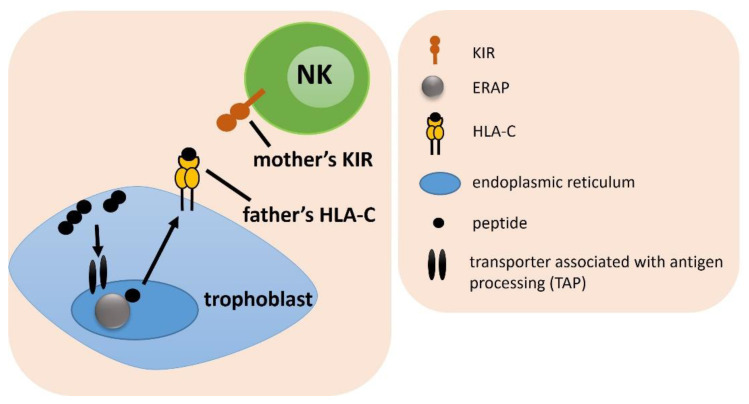
The influence of a female *KIR* and male *HLA-C* on embryo implantation. The arrows indicate the successive stages of the formation of antigenic peptides.

**Figure 2 ijms-23-12518-f002:**
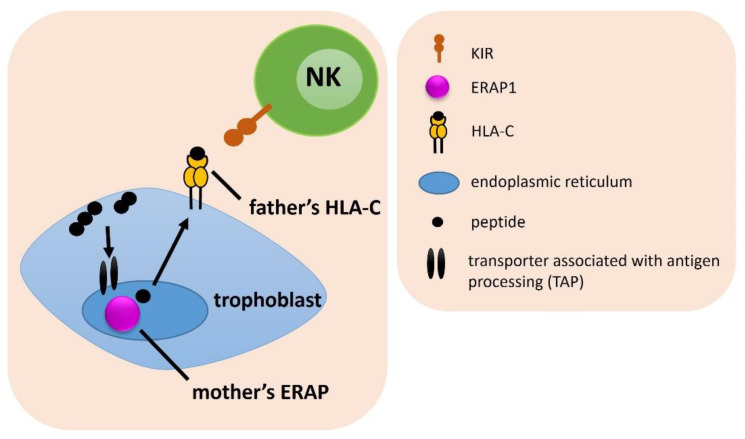
The influence of female *ERAP* and male *HLA-C* on embryo implantation. The arrows indicate the successive stages of the formation of antigenic peptides.

**Figure 3 ijms-23-12518-f003:**
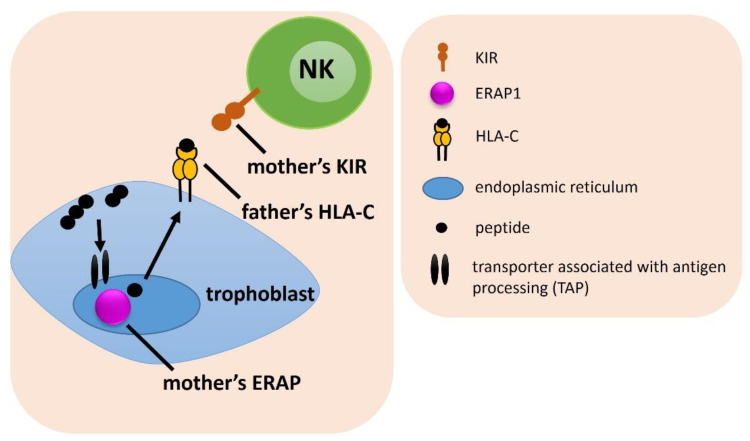
The influence of female *KIR/ERAP* and male *HLA-C* on embryo implantation. The arrows indicate the successive stages of the formation of antigenic peptides.

**Figure 4 ijms-23-12518-f004:**
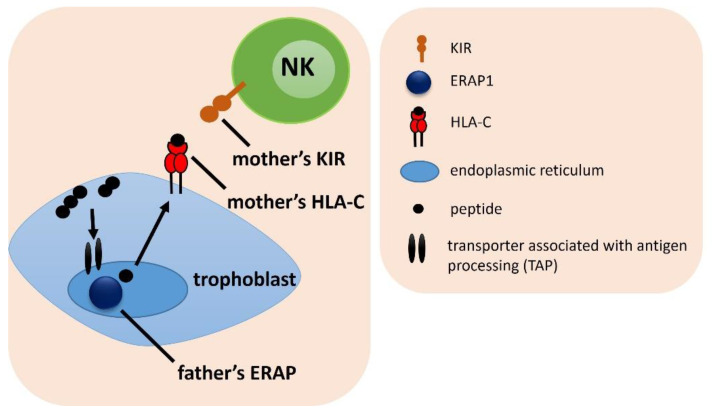
The influence of female *KIR/HLA-C* and male *ERAP* on embryo implantation. The arrows indicate the successive stages of the formation of antigenic peptides.

**Figure 5 ijms-23-12518-f005:**
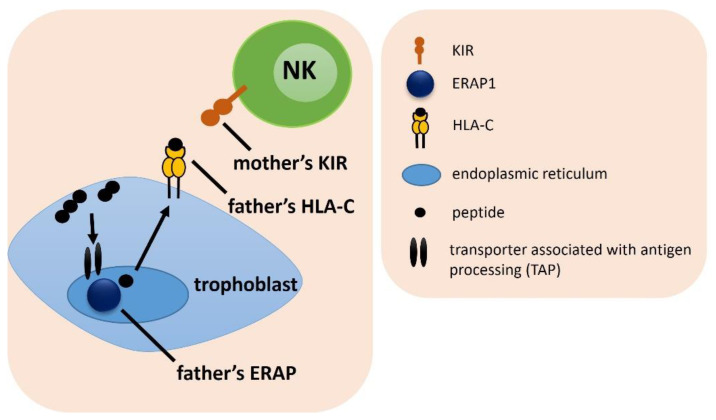
The influence of female *KIR* with male *HLA-C* and *ERAP* on embryo implantation. The arrows indicate the successive stages of the formation of antigenic peptides.

**Figure 6 ijms-23-12518-f006:**
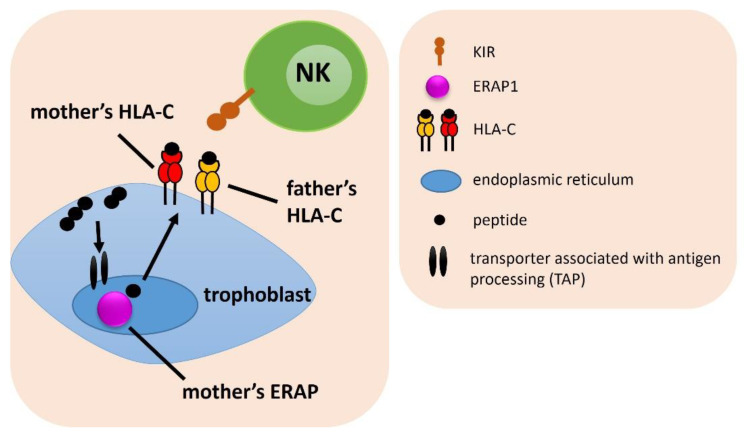
The influence of a mother’s *ERAP/HLA-C* and a father’s *HLA-C* on embryo implantation. The arrows indicate the successive stages of the formation of antigenic peptides.

**Figure 7 ijms-23-12518-f007:**
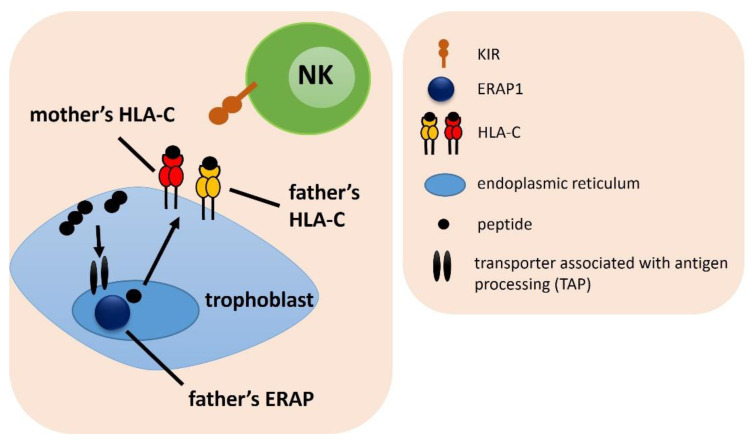
The influence of male *ERAP/HLA-C* and female *HLA-C* on embryo implantation. The arrows indicate the successive stages of the formation of antigenic peptides.

**Figure 8 ijms-23-12518-f008:**
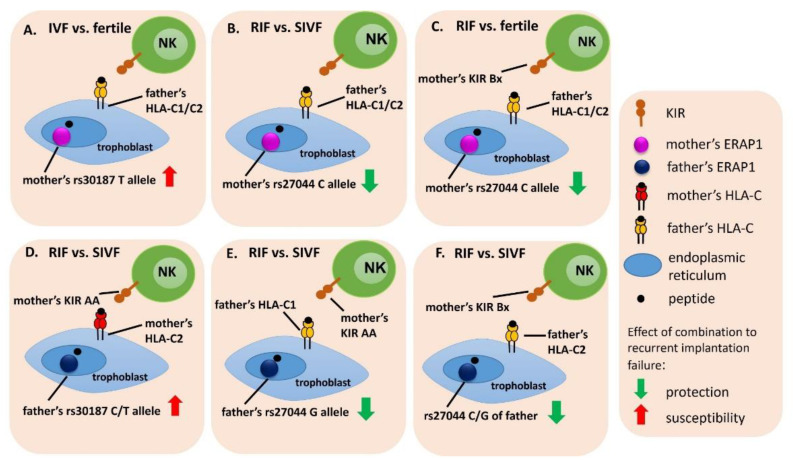
Potential interactions between a woman and her partner in genetic combinations of *KIR/HLA-C/ERAP* and the effect on susceptibility and protection to infertility and recurrent implantation failure (**A**–**F**). IVF means all patients undergoing in vitro fertilization; RIF means patients with recurrent implantation failures; SIVF means patients with successful pregnancy after IVF-ET.

**Table 1 ijms-23-12518-t001:** Summarized effect of *ERAP*, *HLA-C*, *KIR* polymorphisms on susceptibility to infertility and recurrent implantation failure.

ERAP, HLA-C, KIR Combination	Associated Combination	Compared Groups	p	p_corr._	OR	95% CI	Effect	Table
**Female ERAP/male HLA-C**								
ERAP1 rs30187 C>T/HLA-C	TT/C1C2	IVF vs. Fertile	0.006	0.050	2.843	1.30–6.92	↑	Appendix A
ERAP1 rs27044 C>G/HLA-C	CC/C1C1	RIF vs. SIVF	0.005	0.044	0.343	0.15–0.75	↓	Appendix A
ERAP1 rs27044 C>G/HLA-C	CC/C1C2	SIVF vs. Fertile	0.003	0.027	0.442	0.25–0.78	↓	Appendix A
**Female ERAP/male HLA-C/female KIR**								
ERAP1 rs27044 C>G/HLA-C/KIR	CC/C1C2/Bx	RIF vs. Fertile	0.004	0.037	0.428	0.23–0.78	↓	Appendix A
ERAP1 rs27044 C>G/HLA-C/KIR	CG/C1C2/Bx	RIF vs. Fertile	0.006	0.050	2.313	1.25–4.33	↑	Appendix A
**Male ERAP/female HLA-C/female KIR**								
ERAP1 rs30187 C>T/HLA-C/KIR	CT/C2C2/AA	RIF vs. SIVF	0.005	0.049	24.000	1.76–1549.16	↑	Appendix A
**Male ERAP/male HLA-C/female KIR**								
ERAP1 rs27044 C>G/HLA-C/KIR	GG/C1+/AA	RIF vs. SIVF	0.005	0.031	0.114	0.01–0.66	↓	Appendix A
ERAP1 rs27044 C>G/HLA-C/KIR	CG/C2C2/Bx	RIF vs. SIVF	0.004	0.033	0.096	0.01–0.56	↓	Appendix A

↑—susceptibility; ↓—protection; IVF-ET—in vitro fertilization embryo transfer; RIF—recurrent implantation failure; SIVF—successful pregnancy after IVF-ET; p—probability; p_corr._—probability after Bonferroni correction; OR—odds ratio; 95% CI—confidence interval from two-sided Fisher’s exact test.

**Table 2 ijms-23-12518-t002:** Clinical characteristics of patients and fertile couples.

Parameter		IVF	RIF	SIVF	Unclassified	Fertile Control
Number of women		N = 491	N = 278	N = 161	N = 52	N = 322
Age of women	Mean ± SD	33.65 ± 4.13	34.50 ± 4.10	32.11 ± 3.82	34.09 ± 3.91	32.52 ± 5.99
	Range	22–46	23–46	22–41	25–45	22–68
Number of men		N = 491	N = 278	N = 161	N = 52	N = 322
Age of men	Mean ± SD	35.51 ± 4.89	36.27 ± 4.85	34.32 ± 4.72	36.89 ± 4.89	34.15 ± 6.3
	Range	24–53	25–53	24–53	28–51	25–70
Indications for IVF-ET (%)	Only male factor	145 (29.53)	78 (28.06)	58 (36.02)	9 (17.31)	-
	Only female factor	129 (26.27)	68 (24.46)	45 (27.95)	16 (30.77)	-
	Both factors	72 (14.67)	46 (16.55)	19 (11.80)	7 (13.46)	-
	Unknown factor	145 (29.53)	86 (30.93)	39 (24.23)	20 (38.46)	-
Number of IVF-ET	Mean ± SD	3.37 ± 2.06	4.65 ± 1.77	1.63 ± 0.71	1.62 ± 0.68	-
	Range	1–15	3–15	1–3	1–3	-
Number of embryos	Mean ± SD	3.81 ± 2.56	5.35 ± 2.31	1.69 ± 0.79	1.76 ± 0.65	-
	Range	1–19	3–19	1–5	1–3	-

IVF-ET—in vitro fertilization embryo transfer, RIF—recurrent implantation failure, SIVF—successful pregnancy after IVF-ET, SD—standard deviation.

**Table 3 ijms-23-12518-t003:** *ERAP1* and *ERAP2* studied polymorphisms.

SNP	SNP Variation	Amino Acid Change	Gene	Locus	Potential Effect	Assay ID
rs26653	G > C	Pro127Arg	*ERAP1*	5q15	Presumably indirectly affects specificity and enzymatic activity [22]	C__794818_30
rs2287987	T > C	Met349Val	*ERAP1*	5q15	Interactions with the substrate [22]	C__3056893_20
rs30187	C > T	Arg528Lys	*ERAP1*	5q15	Enzymatic activity, expression level [20]	C__3056885_10
rs27044	C > G	Gln730Glu	*ERAP1*	5q15	Enzymatic activity, substrate length preference [22]	C__3056870_10
rs26618	T > C	Ile276Met	*ERAP1*	5q15	Affects efficiency of a precursor peptide trimming for the HLA-C*05-bound epitope [23]	C__3056894_10
rs6861666	A > G	-	*ERAP1/* *ERAP2*	5q15	rs6861666 in 100% LD with rs75862629 which influences the expression level of ERAP2 and ERAP1 [24]	C__29091789_20
rs2248374	A > G	-	*ERAP2*	5q15	Lack of expression of functional forms of the enzyme [20]	C__25649529_10

SNP—single nucleotide polymorphism; LD—linkage disequilibrium.

## Data Availability

The data analyzed in this study is subject to the following licenses/restrictions: data collections can be made available upon request. For now, however, we do not want to make them public due to planned patent solutions. Requests to access these datasets should be directed to KP (karolina.piekarska@hirszfeld.pl).

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
