# Peer review of "ERAP/HLA-C and KIR Genetic Profile in Couples with Recurrent Implantation Failure"

_ijms, 2022, doi:10.3390/ijms232012518_

Round 1

Reviewer 1 Report

Good evening and congratulations on your work!

I find the topic very interesting and applicable as a bench to bedside tool in our practice as ART-IVF specialists. 

I find the introduction very well documented and the topic is easy to follow even to those not familiar to immunologic and genetic causes of RIF. However, I think the design of the study could be improved by adding a subchapter of conclusions after the discussions or maybe further elaborating the discussions. Very well made and self explanatory pictures and charts!!

Author Response

Response to Reviewer 1

We would like to thank the Reviewer for comments and suggestions. We added the last paragraph "Conclusions" after the Materials and Methods.

Reviewer 2 Report

Previous studies have reported HLA-C, KIR and ERAP on maternal side and on the paternal side independently. This is the first manuscript looking at both and provided very important and useful information.

Only caveat that I have and would recommend changing are the figure legends for the schematic drawings. I would provide or name them with the final conclusion from the genetic studies how each combination affects the implantation success or failure. I think this would make the reader not search for the answer in the dense reading of all different genotype association.

Also, KIR on the NK cells is always going to be maternal (keep the same color is acceptable), but the HLA-C combination changes to either maternal or paternal, therefore, I would use different colors to distinguish them. The word labels help but the color association will help and improve faster recognition of the differences.

And the ERAP has two different colors. Does this suppose to represent dimers of ERAP1 and ERAP2? does it ever form a herterodimer of maternal ERAP1 or 2 to paternal ERAP1 or 2? how would you know? and are these compared?

Lastly, I am assuming the full term successful pregnancy did not have any other pregnancy related health issues. I think this need to be stated somewhere in the manuscript. I apologize if it is there but I just missed it. I know it is assumed but better to state.

Author Response

Response to Reviewer 2:

Only caveat that I have and would recommend changing are the figure legends for the schematic drawings. I would provide or name them with the final conclusion from the genetic studies how each combination affects the implantation success or failure. I think this would make the reader not search for the answer in the dense reading of all different genotype association.

We would like to thank the Reviewer for comments and suggestions. Each schematic figure in Results explain visually what combination is considered. Whereas the final conclusion is on the Figure 8 in section Discussion. We improved the all Figures. We changed colors and we added in the legend of Figure 8 explanation of arrows, that this is the effect of combination to recurrent implantation failure. Moreover, we separated the subchapter of Conclusions to emphasize the most important results from this study.

Also, KIR on the NK cells is always going to be maternal (keep the same color is acceptable), but the HLA-C combination changes to either maternal or paternal, therefore, I would use different colors to distinguish them. The word labels help but the color association will help and improve faster recognition of the differences.

We agree with this remark. In all Figures we changed colors the HLA-C and KIR.

And the ERAP has two different colors. Does this suppose to represent dimers of ERAP1 and ERAP2? does it ever form a herterodimer of maternal ERAP1 or 2 to paternal ERAP1 or 2? how would you know? and are these compared?

ERAP in all Figures was originally meant to symbolize ERAP1 and ERAP2. ERAP1 and ERAP2 can form heterodimers but presented pictures are really focused on ERAP1, because we received significant results related to ERAP1. We changed Figures, so that they only show female or male ERAP1.

Lastly, I am assuming the full term successful pregnancy did not have any other pregnancy related health issues. I think this need to be stated somewhere in the manuscript. I apologize if it is there but I just missed it. I know it is assumed but better to state.

We added the information about women in fertile control, that they did not have any health issues in line 459 in the section Materials and Methods.